# Influence of the At-Arrival Host Transcriptome on Bovine Respiratory Disease Incidence during Backgrounding

**DOI:** 10.3390/vetsci10030211

**Published:** 2023-03-10

**Authors:** Mollie M. Green, Amelia R. Woolums, Brandi B. Karisch, Kelsey M. Harvey, Sarah F. Capik, Matthew A. Scott

**Affiliations:** 1Department of Agricultural Sciences, West Texas A&M University, Canyon, TX 79016, USA; 2Department of Pathobiology and Population Medicine, Mississippi State University, Mississippi State, MS 39762, USA; 3Department of Animal and Dairy Sciences, Mississippi State University, Starkville, MS 39762, USA; 4Prairie Research Unit, Mississippi State University, Prairie, MS 39756, USA; 5Tumbleweed Veterinary Services, PLLC, Amarillo, TX 79159, USA; 6Veterinary Education, Research, and Outreach Center, Texas A&M University, Canyon, TX 79016, USA

**Keywords:** bovine respiratory disease, cattle, collagen, cytokine, management, marketing, platelet, RNA, transcriptome, virus

## Abstract

**Simple Summary:**

Bovine respiratory disease is the leading cause of antibiotic use in beef cattle. While several management strategies exist to help prevent bovine respiratory disease, rates of disease remain high, and we do not understand how management decisions, such as marketing decisions, influence the immune and metabolic systems of cattle, especially those related to the development of bovine respiratory disease. Therefore, we evaluated the influence of two different marketing decisions, namely commercial auctioning and direct transportation, and the relationship these decisions have with bovine respiratory disease development on the animal’s immune and metabolic responses, measured through blood RNA sequencing on arrival at a backgrounding operation. We found that cattle that experienced a commercial auction setting prior to arrival displayed gene expression related to increased viral defense, decreased cellular growth and metabolism, and increased innate immunity compared to directly shipped cattle. Individuals who remained clinically healthy during the backgrounding phase after experiencing an auction setting demonstrated increased gene expression related to collagen formation and platelet activity compared with cattle that eventually developed bovine respiratory disease. These results improve our understanding regarding why some cattle develop bovine respiratory disease and establish a basis for future research to maximize beneficial responses in cattle at risk for bovine respiratory disease.

**Abstract:**

Bovine respiratory disease (BRD) remains the leading disease within the U.S. beef cattle industry. Marketing decisions made prior to backgrounding may shift BRD incidence into a different phase of production, and the importance of host gene expression on BRD incidence as it relates to marketing strategy is poorly understood. Our objective was to compare the influence of marketing on host transcriptomes measured on arrival at a backgrounding facility on the subsequent probability of being treated for BRD during a 45-day backgrounding phase. This study, through RNA-Seq analysis of blood samples collected on arrival, evaluated gene expression differences between cattle which experienced a commercial auction setting (AUCTION) versus cattle directly shipped to backgrounding from the cow–calf phase (DIRECT); further analyses were conducted to determine differentially expressed genes (DEGs) between cattle which remained clinically healthy during backgrounding (HEALTHY) versus those that required treatment for clinical BRD within 45 days of arrival (BRD). A profound difference in DEGs (n = 2961) was identified between AUCTION cattle compared to DIRECT cattle, regardless of BRD development; these DEGs encoded for proteins involved in antiviral defense (increased in AUCTION), cell growth regulation (decreased in AUCTION), and inflammatory mediation (decreased in AUCTION). Nine and four DEGs were identified between BRD and HEALTHY cohorts in the AUCTION and DIRECT groups, respectively; DEGs between disease cohorts in the AUCTION group encoded for proteins involved in collagen synthesis and platelet aggregation (increased in HEALTHY). Our work demonstrates the clear influence marketing has on host expression and identified genes and mechanisms which may predict BRD risk.

## 1. Introduction

Bovine respiratory disease (BRD) continues to be one of the leading disease complexes in cattle production in terms of cost, morbidity, and mortality. A previous USDA NAHMS surveillance study in 2011 on U.S. feedlots estimated that BRD costs producers $23.60 per treated animal, with over 16% of cattle placed in feedlots requiring at least one antimicrobial treatment for BRD [1,2]. While BRD has been a priority area of health and disease research for the past several decades, it remains a persistent issue within the U.S. beef industry, with recent studies suggesting a worsening rate of morbidity despite advancements in management schemes, vaccination and therapeutic technologies, and field-level diagnostics [3,4,5]. Consequently, the highly dynamic biology related to BRD and inconsistencies in beef management systems across the U.S. creates difficulties in early detection and risk assessment [6,7,8,9].

Bovine respiratory disease is often defined as an undifferentiated respiratory disease complex [10]. This is primarily due to the multifaceted biology of BRD, including pathogenic interactions, immunological response, and environmental conditions that impact the host [11,12]. Prominently, the beef cattle industry uses the nature and associated risk of these independent components, such as prior animal history, which includes previous administration of vaccines, sale and purchasing records, and commingling and sourcing status, to create a broad categorization of cattle populations into relative risk groups [9,13]. Moreover, beef cattle producers often add value to calves by maintaining health records, adding weight prior to sale to a feedlot, improving uniformity within marketed groups, and preconditioning calves for a feedlot setting [5,14,15]. However, rates of BRD across these broad risk categories are highly variable, and the influence that management decisions such as marketing strategy on host immunity on later performance and BRD incidence is poorly understood.

To better understand the influence that marketing strategy has on host metabolism, inflammation, and immunity, and to determine the ability of host genomic features to predict BRD risk and development, we evaluated at-arrival whole blood transcriptomes of newly weaned cattle that had experienced a commercial auction and order-buyer system prior to backgrounding or that were directly shipped from the cow–calf phase to backgrounding. Our primary objectives were to identify expressed genes and associated biological mechanisms which may distinguish cattle that ultimately develop BRD and to explore whether we could identify the prior marketing history of individuals at backgrounding arrival. Our hypothesis was that gene expression patterns and identifiable associated mechanisms at arrival would identify cattle which would develop BRD during a 45-day period of backgrounding, and that gene expression on arrival may be leveraged to distinguish individuals derived from a commercial auction setting from those directly transported.

## 2. Materials and Methods

### 2.1. Animal Use and Study Enrollment

All animal use and procedures were approved by the Mississippi State University and West Texas A&M University Animal Care and Use Committee (IACUC protocols #19-169 and #2019.04.002, respectively). This study was carried out in accordance with the Animal Research: Reporting of In Vivo Experiments (ARRIVE) guidelines [16]. Eighty-four commercial cross-bred beef steers were randomly assigned into a whole-plot, split-plot design study to evaluate the effect of modified live viral (MLV) vaccination and commercial marketing strategy on health and performance. At the end of the calving season, cattle were selected for subcutaneous MLV vaccination and booster (2 mL SQ; Pyramid 5 (Boehringer Ingelheim Animal Health)) during their cow–calf production phase in Mississippi (Prairie Research Unit, Prairie, MS, USA). Briefly, individuals were housed in six grass-lot pastures (n = 14 cow–calf pairs per pasture), grouped based on vaccination status (VAX (n = 3) or NOVAX (n = 3)) for a median time of 217 days. Pasture groups contained some cow–calf pairs (n = 6 to 9 per group) not enrolled in the study; non-study calves in each pasture group received the same vaccination or not as study calves. At the end of the calving season, cattle were assigned to split-plot level treatment where they were either weaned and housed at the Prairie Research Unit for three days prior to direct shipment to Texas (Texas A&M AgriLife Bushland Research Feedlot, Bushland, TX, USA) (DIRECT, n = 7 calves per pasture group) or transported to a commercial auction market in north Mississippi, housed in pens for approximately six hours, then transported to a regional order-buyer facility for three days, and finally transported to Texas (AUCTION, n = 7 calves per pasture group). Between enrollment and transport to Texas, three calves were removed from the study: one calf from a VAX/AUCTION group was found acutely dead in the pasture approximately 6 weeks before weaning; necropsy revealed a colonic tear and hemoabdomen, likely due to trauma. A second calf from the same VAX/AUCTION group was removed from the study at weaning because it was much smaller (93.6 kg) than all other calves in its group (average weight 223.7 kg), and we concluded that the calf was at significant risk for injury if shipped with larger calves. The third calf, from a different VAX/AUCTION group, was removed from the study at weaning due to a chronic joint injury causing lameness that we determined also increased risk of injury to the calf during transport. These removals resulted in 81 study calves being transported to Texas. Cattle in DIRECT and AUCTION groups were transported to Texas on the same truck but in different compartments, with no contact between groups allowed. Upon arrival in Texas, whole blood was collected from all 81 steers (mean = 235.9 kg, s.d. = 35.6 kg) via jugular venipuncture into Tempus Blood RNA tubes (Applied Biosystems). Cattle were then placed into one of twelve predetermined pens (n = 7 per pen), sorted based on vaccination and sale type status. All cattle were monitored daily for signs of clinical BRD over a 45-day backgrounding period by the same trained observer (SFC); all researchers and trained staff in Texas were blinded to treatment (vaccination, marketing strategy) during data collection. Cattle were assigned a clinical BRD score of 0–4 based on visual signs of disease (Appendix A). Cattle were considered BRD-positive and clinically treated if given a clinical score of 1 or 2 and a rectal temperature >40 °C, or if they were scored a 3 or 4, regardless of rectal temperature. At-arrival samples from all cattle having been treated for clinical BRD after arrival (n = 32) were prioritized for RNA sequencing, and randomly selected samples from clinically healthy cattle (n = 12) with equal proportion across marketing strategies were utilized (n = 6, DIRECT; n = 6, AUCTION). Cattle having been diagnosed with BRD were further categorized into marketing strategy groups (DIRECT, n = 20; AUCTION, n = 12). The overall median time to first treatment was 35 days, with BRD cattle within the AUCTION and DIRECT groups possessing median time to first treatment of 31 and 38 days, respectively. Information for all selected cattle is found in Appendix A.

### 2.2. Sample Processing, Next-Generation RNA Sequencing, and Bioinformatic Processing

Total blood RNA isolation, nucleic acid quality control, Stranded mRNA sequencing library preparation (Illumina, San Diego, CA, USA), and high-throughput shotgun sequencing was performed at the Texas A&M University Institute for Genome Sciences and Society (TIGSS; College Station, TX, USA), in conjunction with our previous work [17]. Total RNA extraction was performed with Tempus Spin RNA Isolation Kits (Thermo Fisher Scientific; Waltham, MA, USA), following the manufacturer’s instructions. Following extraction, total RNA from each sample was then analyzed for concentration and integrity with a Qubit 2.0 Fluorometer (ThermoFisher, Waltham, MA, USA) and an Agilent 2200 Bioanalyzer (Agilent, Santa Clara, CA, USA), respectively; RNA samples were of high quality (RIN: 8.3–9.2; mean = 8.8, s.d. = 0.2) and concentration (ng/μL: 6.4–284.0; mean = 191.3, s.d. = 58.7), with the exception of one sample (S.049.J009), from which we failed to extract RNA. Library preparation for mRNA was completed with the Stranded mRNA Prep Kit (Illumina), following the manufacturer’s instructions. Paired-end sequencing for 150-base-pair read fragments was subsequently performed on an Illumina NovaSeq 6000 analyzer (v1.7+; S4 reagent kit v1.5) in one flow cell lane.

Quality assessment of reads was performed with FastQC v0.11.9 (https://www.bioinformatics.babraham.ac.uk/projects/fastqc/, accessed on 11 April 2022) and MultiQC v1.12 [18], and read-pair trimming for adaptors, undetermined base calling, and retained minimum read length of 28 bases were performed with Trimmomatic v0.39 [19]. Trimmed reads were then mapped and indexed to the bovine reference genome assembly ARS-UCD1.2 with HISAT2 v2.2.1 [20]. Sequence Alignment/Map (SAM) files were converted to Binary Alignment Map (BAM) files prior to transcript assembly via Samtools v1.14 [21]. Transcript assembly and gene-level expression estimation for differential expression analysis was performed with StringTie v2.1.7 [22], as described by Pertea and colleagues [23]. Three samples (S.054.J017, S.087.J113, and S.090.J123) were considered of low quality and technical outliers per the initial quality control assessment and were subsequently removed from further analysis. All raw sequencing data produced in this study are available at the National Center for Biotechnology Information Gene Expression Omnibus (NCBI-GEO) under the accession number GSE218061.

### 2.3. Differential Gene Expression and Functional Enrichment Analyses

Gene-level raw count matrices were explored within RStudio, via R v4.1.2. Raw counts were processed and filtered by procedures previously described [17,24]. Retained data were normalized for differential expression analysis with the Trimmed Mean of M-values method (TMM) [25]. Mixed effect statistical modeling was performed with edgeR v3.36.0 generalized linear model likelihood ratio testing (glmLRT), following tagwise dispersion estimate fitting. Specifically, analyses were performed in four steps: (1) HEALTHY (n = 10) versus BRD (n = 30) across all cattle (blocking for sale type), (2) AUCTION (n = 16) versus DIRECT (n = 24), (3) HEALTHY (n = 5) versus BRD (n = 11) within the AUCTION group, and (4) HEALTHY (n = 5) versus BRD (n = 19) within the DIRECT group; all testing was performed with additive models, accounting for vaccination (VAX) and pen order (MS_pasture) from when cattle were raised in Mississippi prior to transport to Texas. Genes were considered differentially expressed with a false discovery rate (FDR) of less than 0.05. Visual relationships of the genes identified by each analysis was performed with UpSetR v1.4.0 [26], utilizing the interactive interface Intervene [27].

Identified differentially expressed genes (DEGs) were analyzed for enrichment of gene ontology terms, including biological processes, molecular functions, and cellular components, and pathways via the Reactome pathway database [28] and over-representation analysis through WebGestalt 2019 (WEB-based GEne SeT AnaLysis Toolkit) API [29]. Over-representation analysis parameters within WebGestalt 2019 included the *Bos taurus* genome as the reference set, between 2 and 2000 genes per category, Benjamini–Hochberg (BH) procedure for multiple hypothesis correction, FDR cutoff of 0.05 for significance, and a total of 10 expected reduced sets of the weighted set cover algorithm for redundancy reduction. Enriched gene ontology terms, specifically biological processes, and the Reactome pathways were evaluated for their directionality (increased or decreased) based on log2 fold changes of associated DEGs.

### 2.4. Data Visualization and Model-Based Unsupervised Clustering Analyses

To reduce the high dimensionality of the gene expression dataset and to identify potential correlations with clinical metadata (Appendix A), principal component analysis (PCA) was performed with the Bioconductor package PCAtools v2.10.0 (https://github.com/kevinblighe/PCAtools, accessed on 11 November 2022), utilizing a correlation matrix. Correlation matrix modeling was selected due to the uneven scale of variations generally found within gene expression data, where covariance matrix modeling tends to be less informative due to the skewness by the most variable and/or lowest abundant genes [30,31,32]. Trimmed Mean of M-values normalized gene expression counts were log2-transformed after the addition of a (+1) pseudocount to prevent log-transformation of any zero counts, processed with mean-centering (“center = TRUE”) and variance-scaling (“scale = TRUE”), and the bottom 10% of genes with the lowest total variance across samples (“removeVar = 0.1”) were removed. A scree plot was used to determine the number of principal components (PCs) to be retained for further analysis, employing the Elbow and Horn’s parallel analysis methods, respectively [33]. Spearman’s rank correlations of the retained PCs were calculated with metadata components across all samples, specifically for average daily weight gain from birth until allocation to sale type (MSADG), age (in days) at time of Texas arrival (ArrivalAge), shrunk weight upon Texas arrival (ArrivalWt), binary coding if an individual was administered two modified live viral respiratory vaccines during the cow–calf phase in Mississippi (Vaccination; 0 = No, 1 = Yes), days at risk for BRD development during the backgrounding phase in Texas (Risk; maximum = 45 d), at-arrival fecal parasite egg counts per gram of feces calculated via the Modified McMaster technique on the same day of collection (EPG), the 25-acre pasture identity where individuals were housed in Mississippi during the cow–calf phase (Pasture), binary coding for the type of sale system the individual moved through prior to Texas arrival (Sale; 0 = AUCTION, 1 = DIRECT), the number of clinical treatments an individual received for BRD throughout the Texas backgrounding phase (Severity; min = 0, max = 2), and if an individual ever received treatment for clinical BRD throughout Texas backgrounding (Disease; 0 = BRD, 1 = Healthy). Spearman’s correlations were considered to have significant associations with an FDR < 0.10. A PCA biplot was then constructed from the first PCs with significant correlations to metadata (PC2 and PC3) with the “encircle = TRUE” function selected to automatically depict a polygon around groups specified by SALE (top correlated feature; PC2). Lastly, to identify which genes were the primary drivers of the variation that was seen in each significantly correlated PC (PC2, PC3, and PC7), a loadings plot was generated with the top/bottom 1% of retained variables across each of the component loading range.

Following PCA, initial gene expression patterns within each analysis dataset (i.e., HEALTHY vs. BRD, AUCTION vs. DIRECT, etc.) were explored by applying multidimensional scaling (MDS) to each gene expression dataset after gene count filtering and TMM normalization, using the plotMDS function from the edgeR package. Visualizing differences in gene expression patterns via MDS is accomplished through unsupervised clustering of the root-mean-square average of the log-fold-changes for selected genes identified in each sample, allowing for the generalization of dissimilarities and potential batch effects within the dataset [34]. Analysis via MDS was performed with “top = 500” to select the top 500 genes ranked on standard deviation for calculating distances, “gene.selection = common” to select the same genes for all comparisons, and “dim.plot = c(1,2)” to plot the first two principal components.

Once DEGs were identified from each analysis, TMM-normalized counts were converted into log2 count-per-million (log2CPM) values for heatmap construction with the Bioconductor package pheatmap v1.0.12 (https://cran.r-project.org/web/packages/pheatmap/index.html, accessed on 11 November 2022). Data-centered and normalized z-scores from log2CPM counts were utilized for depiction and clustering of relative gene-wise variation of gene expression. Pearson correlation coefficients and Euclidean distances were calculated for clustering dissimilarities by column (sample) and row (gene), respectively. Color scaling for data visualization was performed with the Bioconductor package viridis v0.6.2 [35] to allow for ease of visual interpretation for individuals affected with color blindness.

## 3. Results

Post-quality control and read trimming yielded a total of 1,431,561,514 filtered reads across all 43 samples (median = 33,259,429 reads per sample, s.d. = 2,938,551); mapping and alignment of trimmed reads to the *Bos taurus* reference genome assembly (ARS-UCD1.2) resulted in an average overall alignment rate of 95.9% (Appendix A). Post-alignment and gene-count matrix construction resulted in 33,310 unique annotated features. Following pre-processing and count filtering, 16,741 genes were retained for clustering and differential expression analyses. Prior to differential expression analysis, global gene expression patterns were evaluated through PCA. Using both the Horn’s Parallel analysis and Elbow method, the first eight principal components were retained for downstream analysis, which accounted for 57.1% of the total variance (Figure 1A). Spearman’s correlation of PCs with metadata components demonstrated significant correlations with Sale and PC2 (12.0% variance explained; r = −0.82, FDR < 0.01), ArrivalWt and PC3 (6.0% variance explained; r = −0.48, FDR < 0.05), MSADG and PC3 (6.0% variance explained; r = −0.44, FDR < 0.10) and ArrivalAge and PC7 (2.7% variance explained; r = 0.55, FDR < 0.01) (Figure 1B). A biplot of PC2 and PC3 was constructed to visualize the high relative dissimilarity of the samples, demonstrating a distinct pattern between individuals by Sale and no discernable pattern by Disease (Figure 1C). The top eight genes influencing these patterns within each PC (i.e., component loading) were CCDC146, EPSTI1, LOC101906463, LOC112443219, LOC507247, RTP4, SLFN11, and TRIM14 in PC2, and ANKRD34A, BOLA-DQA, CITA, DIRAS2, FFAR3, SKAP2, TMEM145, and TRABD2B in PC3. Those genes determined to be the drivers of variation within each of the three significantly correlated PCs are indicated by the Loadings Plot (Figure 1D). Genes driving variation which was correlated specifically with AUCTION cattle (PC2) included ACOD1, ANKRD34A, ANKRD50, CCDC146, CCNF, CDCA8, DPYD, EPSTI1, LOC101906463, LOC104971363, LOC112442703, LOC112443219, LOC507247, rna-NR_031144.1, RTP4, SKAP2, SLFN11, SPC24, TARM1, TMEM145, and TROAP. Genes driving variation which was negatively correlated with average weight gain in Mississippi and shrunk weight at arrival (PC3) included the aforementioned genes, with the exclusion of LOC507247, rna-NR_031144.1, and SLFN11. The exact genes identified by PC2 were also found to be positively correlated with age at Texas arrival within PC7.

Differential expression analysis of all samples, evaluating HEALTHY versus BRD, resulted in one differentially expressed gene (DEG), BOLA-DQA5, which was decreased on arrival in cattle that remained healthy during backgrounding (Appendix A). As previously described by PCA, Sale possessed a significant influence on the gene expression dataset, and no discernable patterns were observed when evaluating for Disease or Severity. Subsequently, differential expression analysis was performed independently for AUCTION versus DIRECT, HEALTHY versus BRD within the AUCTION group, and HEALTHY versus BRD within the DIRECT group, which resulted in 2961 (1538 increased, 1423 decreased in DIRECT), 9 (all increased in HEALTHY), and 4 DEGs identified (3 increased, 1 decreased in HEALTHY), respectively (Appendix A). Visualization of the number and overlap of each DEG identified by each analysis are found in Figure 2.

Multidimensional scaling (MDS) and heatmap clustering was performed with three specific comparisons: (1) AUCTION versus DIRECT groups (including labeling for disease) (Figure 3), (2) HEALTHY versus BRD within the AUCTION group (Figure 4), and (3) HEALTHY versus BRD within the DIRECT group (Figure 5). Euclidean distances based on the top 500 variable genes of all cattle demonstrated distinct clustering by sale type within the first principal component (*x*-axis) based on MDS, with little separation by disease status (Figure 3A). Further evaluation of the 2961 DEGs identified between sale types via heatmap clustering substantiated the distinction in gene expression patterns between the two sale type groups, with little division between disease status across all samples (Figure 3B). Visualization of gene expression variation within the AUCTION group via MDS demonstrated relative separation of disease groups within the first principal component (*x*-axis), which explained 9% of the total variance within the dataset (Figure 4A). Hierarchical clustering of the nine DEGs identified by disease within the AUCTION group supported the findings of the aforementioned MDS plot, depicting separation of disease groups by the relative expression of these nine genes (Figure 4B). Lastly, visualization of gene expression variation within the DIRECT group via MDS demonstrated no clear separation of disease groups (Figure 5A). Further hierarchical clustering of the four DEGs identified between disease cohorts within the DIRECT group did demonstrate separation of cattle that eventually developed BRD; however, two of the five analyzed HEALTHY cattle trended with BRD cattle, based on segmentation of the columns (samples) into two based on hierarchical patterns (Figure 5B).

Functional enrichment analysis was separately performed with DEGs identified within (1) AUCTION versus DIRECT and (2) HEALTHY versus BRD within the AUCTION group. Functional enrichment in HEALTHY versus BRD groups across all samples and HEALTHY versus BRD within the DIRECT group could not be performed due to there being too few genes (n = 1 and n = 4, respectively). Analysis of the DEGs identified from AUCTION versus DIRECT revealed enrichment for 113 biological processes, 44 cellular components, 4 molecular functions, and 54 Reactome pathways (Appendix A). Biological processes identified within AUCTION versus DIRECT were related to the innate immune response (increased in AUCTION), protein metabolism and secretion (increased in AUCTION), viral response and type I interferon production (increased in AUCTION), response to interferon gamma (increased in AUCTION), response to external stimuli and cytokines (decreased in AUCTION), autophagy (increased in AUCTION), response to bacteria (increased in AUCTION), and fatty acid mobilization and metabolism (decreased in AUCTION). Cellular components identified from DEGs between AUCTION and DIRECT cattle involved the cytosol, nuclear lumen and nucleoplasm, both extracellular and intracellular vesicles, lipid droplets, ribosomes, and mRNA-editing complexes. Molecular functions identified between AUCTION and DIRECT cattle included structural constituent of ribosomes, pattern recognition receptor activity, and anion binding. The Reactome pathways enriched between AUCTION and DIRECT cattle included neutrophil degranulation (decreased in AUCTION); antiviral mechanisms by interferon, including ISG15-mediated antiviral activity (increased in AUCTION); antigen processing and cross-presentation, including MHC class I-mediated processing and presentation (increased in AUCTION); B-cell receptor signaling and activation of NF-κB (increased in AUCTION); MyD88-independent and TRIF-mediated toll-like receptor 4 signaling (increased in AUCTION); and p53-independent DNA damage response (increased in AUCTION).

Analysis of the DEGs identified from HEALTHY versus BRD within the AUCTION group revealed enrichment for 2 biological pathways, 8 cellular components, 2 molecular functions, and 21 Reactome pathways (Appendix A). Biological processes identified were protein heterodimerization (increased in HEALTHY) and skin morphogenesis (increased in HEALTHY). Cellular components identified were collagen type I and fibrillar collagen trimers, banded collagen fibril, and collagen-containing extracellular matrix components. Molecular functions identified were related to extracellular matrix structural constituents and platelet-derived growth factor binding. Reactome pathways identified were related to extracellular cellular matrix proteoglycans (increased in HEALTHY), platelet activation/aggregation and adhesion to exposed collagen (increased in HEALTHY), GP1b-IX-V activation signaling (increased in HEALTHY), collagen biosynthesis and formation (increased in HEALTHY), and immunoregulatory interactions between lymphoid and non-lymphoid cells (increased in HEALTHY).

## 4. Discussion

Broadly, research has demonstrated that cattle sold through a commercial auction setting and placed in novel commingling settings tend to be classified at higher risk for BRD development [6,14]. However, it is often difficult to separate out the impact of some of the other BRD-related risk factors that may accompany cattle marketed via an auction (lack of preconditioning, commingling, stress, social group disruption, pathogen exposure, abrupt or high stress weaning, lack of proper prior nutrition, unknown immunological status, etc.) [6,9,13,36,37]. Additionally, although we have identified some BRD-related risk factors, their exact direct influence on BRD development and the combined additive or multiplicative interactions among risk factors are relatively unknown and can be highly variable, as not every animal who moves through an auction has the same underlying experiences. The group of cattle evaluated in this study is unique in that their whole life history was known and every aspect of their management was planned and meticulously followed from the time of dam insemination to the end of backgrounding. This provided us a unique and invaluable opportunity to study the impact of marketing decisions without any potential confounding or effect modifying factors and to account for other factors.

Even so, the conditions that we raised these cattle under were not inclusive of all ways cattle are raised and all risk factors that cattle may experience prior to backgrounding. For our cattle, we controlled for prior vaccination with a modified live viral respiratory vaccine. We castrated all calves 69 days prior to abruptly weaning all animals. The type of auction market exposure we designed included a relatively short course to a local auction market, a short (~6 h) stay there as a group where they could have fence line contact with other cattle, and then a brief stay at a local order-buyer facility, where they remained for three days prior to shipment to Texas. Within the realm of “auction market” systems, there is certainly much variation in the time it takes to transport cattle to a market, e.g., the length of time animals spend at the actual market and how much or little they are commingled or exposed to pathogens there, the time and distance to their next destination, etc., that could result in more or less stress and risk of subsequent disease. Similarly, even cattle who are directly transported to the next phase in the production cycle experience variations in prior management and transport time, distance, and conditions that may produce variation in outcomes. Therefore, our results may not be applicable to all cattle that move through a market system or that are directly transported; thus, further research exploring the variability in management and marketing is needed.

Our objective in this study was to identify differences in gene expression and associated host-driven biological systems that may be impacted by marketing decisions and how these influence eventual BRD morbidity after backgrounding arrival, in order to explore potential mechanisms of BRD development or resistance that may be leveraged in future studies. While both our research group and others have used the blood RNA-Seq methodology to identify potential predictive markers and mechanisms related to BRD [38,39,40,41,42,43,44,45], this study, to our knowledge, is one of the first to utilize said technologies to evaluate how marketing decisions, with relationship to BRD, influence inflammatory- and immune-mediated mechanisms in cattle. Importantly, exposure to an auction market setting and an order-buyer facility for only three days was associated with the differential expression of 2961 genes representing 113 biological processes. This striking difference in gene expression between the two groups of cattle that originated from the same herd illuminates the numerous immunologic and metabolic processes that can be affected by exposure to a marketing environment. It is notable that the cattle were never physically mixed with other cattle during auction market exposure, so commingling of cattle from outside sources was not a factor in the changes in gene expression observed.

Our initial results evaluating HEALTHY and BRD cattle at arrival yielded one DEG, BOLA-DQA5, that was decreased in HEALTHY cattle relative to BRD cattle. Members of the major histocompatibility complex class IIa region, of which BOLA-DQA5 is part of, have been researched in association with cattle diseases, such as viral infection and mastitis [46,47,48]. More recently, BOLA-DQA5 was a genotype target for genetic architecture research in Holsteins [49,50]; however, its relationship with cattle health and disease development is largely unknown at this time. Moreover, the lack of significant findings associated with overt disease (i.e., between all samples used) can be attributed to the high amount of variance explained by marketing decision alone, shown in our dimensional reduction analyses (Figure 1 and Figure 3). Therefore, we first investigated what genes and mechanisms were driving this variation. Component loadings from our PCA determined that ACOD1, ANKRD34A, ANKRD50, CCDC146, CCNF, CDCA8, DPYD, EPSTI1, LOC101906463, LOC104971363, LOC112442703, LOC112443219, LOC507247, rna-NR_031144.1, RTP4, SKAP2, SLFN11, SPC24, TARM1, TMEM145, and TROAP were the primary drivers of variation in association with marketing decision. Further evaluation of genes detected in cattle distinguished by marketing decision identified a total of 2961 DEGs, of which all identified component loading genes overlapped. Several of these DEGs, including ACOD1, ANKRD34A, ANKRD50, LOC507247, RTP4, SLFN11, and TARM1, have been shown to be involved in macrophage-directed inflammatory mechanisms and antiviral response, specifically centered around type-I interferon production [51,52,53,54,55,56,57,58,59]. Moreover, our functional enrichment analysis of these DEGs showed that they were largely involved in antiviral defense/type-I interferons (increased in AUCTION); cell growth regulation (decreased in AUCTION); immune activation, centered around toll-like receptor 4 activity and complement activity (increased in AUCTION); and inflammatory mediation and lipid metabolism (decreased in AUCTION). While a limitation of this study is the lack of respiratory metagenomic or viral identity information, our findings suggest that these cattle, only having been placed in a commercial auction setting for a relatively short period of time, were exposed and immunologically responded to a virulent virus or viruses [17,60,61,62,63]. Interestingly, these antiviral-related gene expression signatures were not necessarily associated with clinical BRD development and severity during backgrounding, as seen in previous RNA-Seq studies [39,42,45,64]. Future studies should pair RNA-Seq with host genetic and/or epigenetic evaluation and pathogen or microbiome identification methods to more clearly associate pathogen exposure and regulation with regards to these DEGs. While this finding does not negate the influence that viruses and prolonged inflammatory activity may have on BRD development, these type-I interferon-related gene expression patterns observed at backgrounding arrival may give us the ability to retrospectively identify cattle that may have experienced viral exposure and/or prior auction market exposure and may help us better categorize their risk status. Additional research is needed to see if we can refine this methodology to identify “stale” cattle who have spent more than several days in the marketing system and further pinpoint cattle at higher risk of disease or poor performance during backgrounding or feeding. Furthermore, approximately 40% (32/81) of cattle entering this 45-day backgrounding period were subsequently diagnosed with BRD. While this relatively high rate of BRD is not uncommon in commercial beef production systems, the overall frequency of BRD treatment for this population is higher than cattle populations, especially those of the relatively same age and weight, as shown in our previous work [38,41,64]. Potentially, these cattle, having been maintained and transported from a single-source, relatively low-risk environment, were exposed to pathological and environmental features novel to them, and/or our detection of clinical BRD was more rigorous compared to large commercial operations.

To account for the large amount of variation driven by marketing decisions, we further split the dataset by AUCTION or DIRECT, to identify at-arrival gene expression patterns and mechanisms which may indicate eventual clinical BRD development within each group. Beginning with the AUCTION group, we identified relative separation of disease groups based on expressional variation (Figure 4) and a total of nine DEGs between HEALTHY and BRD. These DEGs primarily are involved in collagen biosynthesis and modification and platelet adhesion and aggregation, which were relatively increased in HEALTHY calves. Recently, Johnston and colleagues discovered that COL1A1 and COL1A2, the genes driving the aforementioned mechanisms, were the most down-regulated genes in whole blood collected from BRSV-challenged calves compared to sham-control calves [63]. While the exact mechanism of how these type-I collagen-associated genes relate to viral exposure and subsequent BRD development is unknown at this time, they are involved in airway macrophage-driven cell clearance, metalloproteinase regulation, and fibrogenesis [65,66,67]. Furthermore, platelet activity is linked to collagen exposure and is shown to increase both the adhesion capacity of lymphocytes and enhance T-cell differentiation [68,69,70]. Collectively, this may serve as a predictor of, and possible protective mechanism in, BRD development in auction-marketed, viral-infected cattle, which warrants future investigation.

Lastly, our analysis of cattle within the DIRECT group yielded no discernable patterns related to BRD outcome (Figure 5), with only four DEGs identified: EFEMP1, HELQ, LOC112445634, and LOC112446743. Due to the low number of DEGs identified, we were unable to ascertain unified functional enrichment within this analysis. To our knowledge, previous research has not identified nor linked these genes to infectious respiratory disease in mammals.

One key feature, and subsequent limitation, of our study was the timing to the first BRD treatment. These cattle possessed an overall median time to first treatment of 35 days, with BRD cattle in the AUCTION and DIRECT groups having median times of approximately 31 and 38 days, respectively (Appendix A). While discernable differences in gene expression were identified within the AUCTION group in relation to BRD development, the overall lack of DEGs identified related to BRD may be attributable to at-arrival gene expression patterns not being capable of representing BRD morbidity when disease occurs over four weeks post-sampling, as was the case for many of the cattle in this study. This is in contrast to the typical pattern of BRD in recently transported cattle, in which disease is expected (2–4 weeks post-arrival) [3,71,72]. Additionally, our study dependently evaluated BRD from one clinical illness scoring system (Appendix A), while several concurrently exist in commercial production systems. In addition, visual assessment alone may not accurately identify all BRD cases. Moreover, not all cattle moving through a commercial auction setting may be exposed to a virus or viruses. As such, future research evaluating the host response with regards to different approaches of identifying and diagnosing BRD (e.g., lung ultrasonography, cytological evaluation of the airway, etc.), as well as the relationship with microbial exposure and/or upper respiratory microbiota, is imperative.

## 5. Conclusions

This study was conducted to explore whole blood gene expression profiles of newly received cattle at a backgrounding operation in order to determine patterns and specific genes and genomic mechanisms related to marketing decisions and BRD development during backgrounding. Here, we describe nearly 3000 differentially expressed genes with a distinction between cattle processed through a commercial auction setting compared with cattle directly shipped to backgrounding; these DEGs are hallmarked by genes related to type-I interferon production, toll-like receptor 4 activity, cell growth regulation, and lipid metabolism. While the prolonged time to BRD incidence may have influenced our inability to capture information related to the influence of marketing decisions on the diagnosis and treatment of BRD, key differences related to collagen formation and metabolism were identified within auctioned cattle that resisted or developed clinical BRD. These results, in accompaniment with the findings of previous RNA-Seq research, provide new information about gene expression pathways activated by the process of auction market exposure. These results contribute to a growing body of knowledge regarding gene expression pathways related to management practices and BRD risk.

## Figures and Tables

**Figure 1 vetsci-10-00211-f001:**
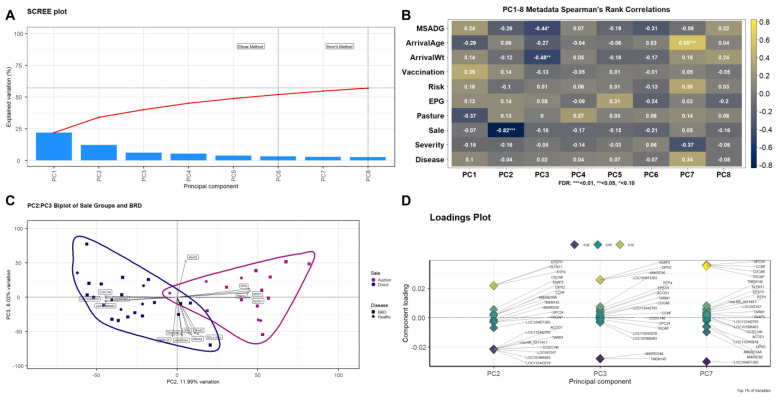
Principal component analysis (PCA) of the global gene expression data generated for all 40 samples utilized. (**A**) Scree plot depicting the first eight PCs retained for further PCA, which described over 57% of the total explainable variance. (**B**) Heatmap of Spearman’s Rank correlation coefficients that were associated with metadata components within each of the eight retained PCs. Clinical metadata components are, in descending order, average daily weight gain from birth until allocation to sale type (MSADG), age (in days) at time of Texas arrival (ArrivalAge), shrunk weight upon Texas arrival (ArrivalWt), binary coding if an individual was administered two modified live viral respiratory vaccines during the cow–calf phase in Mississippi (Vaccination; 0 = No, 1 = Yes), days at risk for BRD development during the backgrounding phase in Texas (Risk; maximum = 45 d), at-arrival fecal parasite egg counts per gram of feces calculated via Modified McMaster technique on same day of collection (EPG), the 25-acre pen identity where individuals were housed in Mississippi during the cow–calf phase (Pasture), binary coding for the type of sale system the individual moved through prior to Texas arrival (Sale; 0 = AUCTION, 1 = DIRECT), the number of clinical treatments an individual received for BRD throughout the Texas backgrounding phase (Severity; min = 0, max = 2), and if an individual ever received treatment for clinical BRD throughout Texas backgrounding (Disease; 0 = BRD, 1 = Healthy). Color represents the R-value identified between each PC and metadata component; yellow/white cells represent a higher positive value, purple/black cells represent a lower negative value. Significance was calculated through FDR adjustments and is indicated by * FDR < 0.10, ** FDR < 0.05, or *** FDR < 0.01. (**C**) A biplot of PC2 and PC3, where samples were colored by sale type (purple = AUCTION, blue = DIRECT) and shaped by disease (square = BRD, diamond = Healthy). Individual plots (vectors) represent the PC score of the individual sample by gene expression, and vector distances along the *x*- and *y*-axes represent the total variational influence. Genes driving the explained variance for each PC are represented by arrows (directionality) and name. (**D**) Loadings plot with annotated genes driving associated variation and directionality (*y*-axis) of PC2, PC3, and PC7. The top 1% of genes identified by variance are seen as the most responsible for driving variation with each of the aforementioned PCs. Color (dark yellow to dark blue; positive to negative) demonstrates the corresponding directionality of expression and strength of influence for each gene within each PC.

**Figure 2 vetsci-10-00211-f002:**
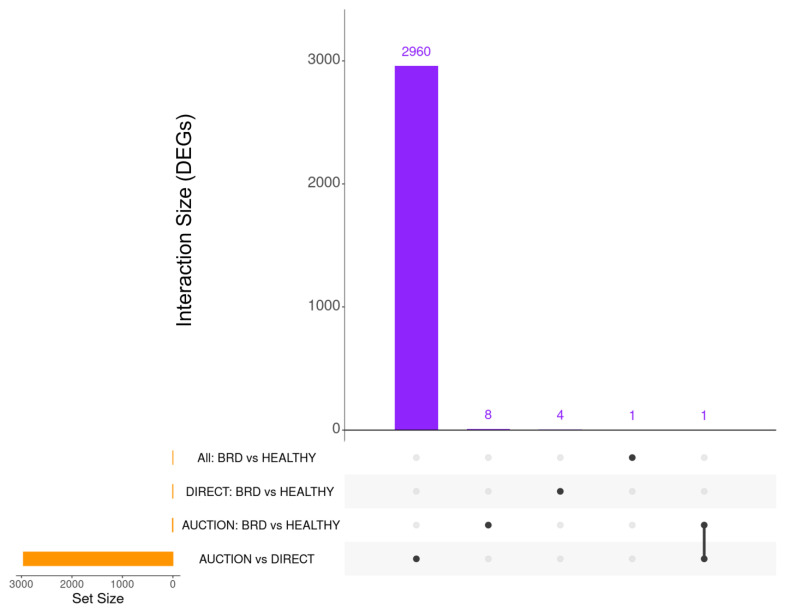
Upset plot representing the total number of DEGs identified by each analysis (Set Size) and the number of DEGs overlapping between analyses (Interaction Size). AUCTION versus DIRECT demonstrated the greatest number of unique DEGs (2960) across all analyses, with only one overlapping gene identified between any of the four analyses (RN18S1; AUCTION versus DIRECT and BRD versus HEALTHY in AUCTION cattle).

**Figure 3 vetsci-10-00211-f003:**
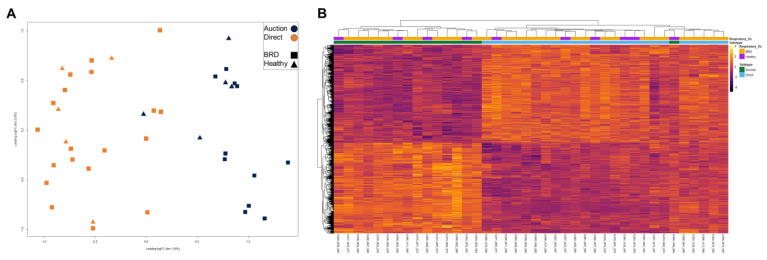
Multidimensional scaling (MDS) and unsupervised hierarchical clustering of gene expression between all AUCTION and DIRECT cattle at facility arrival. (**A**) Points within the MDS plot represent each sample and their transformed Euclidean distance in the first two principal components, observed as the leading log2-fold change between the common distances of the top 500 genes that best differentiate each sample. (**B**) Heatmap and hierarchical clustering of the 2961 DEGs identified between all AUCTION and DIRECT cattle. Gene expression values were scaled and normalized with z-scores calculated from log2 count-per-million transformed, Trimmed Mean of M-values (TMM)-normalized counts. Samples were labeled according to BRD acquisition (Respiratory_Dz) and method of sale (Saletype). Relative expression of each gene is depicted from high (yellow/white) to low (purple/black) within each sample.

**Figure 4 vetsci-10-00211-f004:**
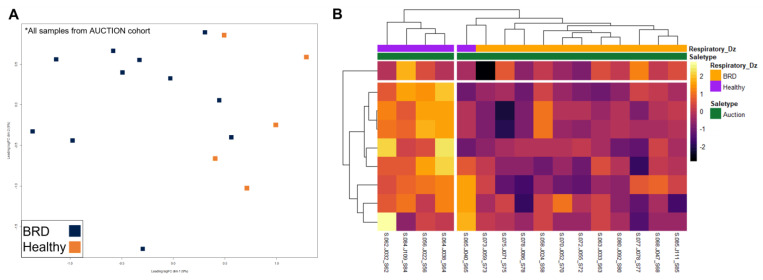
Multidimensional scaling (MDS) and unsupervised hierarchical clustering of gene expression between BRD and HEALTHY cattle within the AUCTION group at backgrounding arrival. (**A**) Points within the MDS plot represent each sample and their transformed Euclidean distance in the first two principal components, observed as the leading log2-fold change between the common distances of the top 500 genes that best differentiate each sample. (**B**) Heatmap and hierarchical clustering of the nine DEGs identified between disease cohorts within the AUCTION group. Gene expression values were scaled and normalized with z-scores calculated from log2 count-per-million transformed, Trimmed Mean of M-values (TMM)-normalized counts. The samples were labeled according to BRD acquisition (Respiratory_Dz). Relative expression of each gene is depicted from high (yellow/white) to low (purple/black) within each sample.

**Figure 5 vetsci-10-00211-f005:**
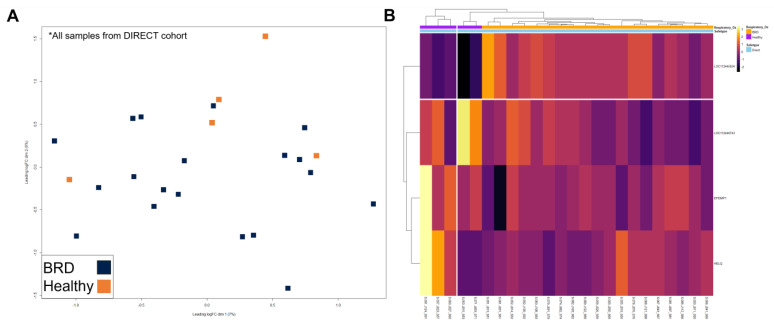
Multidimensional scaling (MDS) and unsupervised hierarchical clustering of gene expression between BRD and HEALTHY cattle within the DIRECT group at facility arrival. (**A**) Points within the MDS plot represent each sample and their transformed Euclidean distance in the first two principal components, observed as the leading log2-fold change between the common distances of the top 500 genes that best differentiate each sample. (**B**) Heatmap and hierarchical clustering of the four DEGs identified between disease cohorts within the DIRECT group. Gene expression values were scaled and normalized with z-scores calculated from log2 count-per-million transformed, Trimmed Mean of M-values (TMM)-normalized counts. The samples were labeled according to BRD acquisition (Respiratory_Dz). Relative expression of each gene is depicted from high (yellow/white) to low (purple/black) within each sample.

## Data Availability

All relevant data are within the paper and its Appendix A. All raw sequencing data produced in this study are available at the National Center for Biotechnology Information Gene Expression Omnibus (NCBI-GEO), under the accession number GSE218061.

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
