# Peer review of "Influence of the At-Arrival Host Transcriptome on Bovine Respiratory Disease Incidence during Backgrounding"

_vetsci, 2023, doi:10.3390/vetsci10030211_

Round 1

Reviewer 1 Report

The manuscript by Green et al. is focused on whole blood transcriptome changes in calves that go through auction vs. direct shipped, and those that develop BRD vs. not.  The manuscript is well written and the data are interesting related to marketing impacts on the transcriptome.  I think some additional detail related to the timing of the BRD events should be included in the results section.  The authors do not comment on the timing of the mean time to the first BRD events and I think this is an important component of the results that should be expanded on.  Previous studies by this group of seen BRD events that happened much sooner following cattle placement.  I think that the authors should also consider some discussion on the overall frequency of the BRD events in this group of calves.  A very high frequency of animals were diagnosed as BRD. While it is not unheard of, the high disease prevalence is also different than previous work from this group. Some discussion about this population of calves as a whole - and why the high disease prevalence - would be something to consider.

Author Response

  • In regard to timing of BRD diagnosis, our manuscript includes lines 144-146 and 555-567, which detail the median timing and limitation of BRD diagnosis in this study, and Supplemental File S2 for all relevant clinical details. More specifically, on lines 506-522, our at-arrival gene expression findings related to commercially auctioned cattle are highly similar to previous RNA-Seq studies (type I interferon-related gene expression); however, those previous studies found that increased type I interferon-related gene expression was related to disease outcomes, whereas we identified this only in relation to marketing strategy and not BRD. A limitation of this study, and thus future direction, is the lack of pathogen identification (metagenomic evaluation, viral isolation, etc.), which may further categorize those cattle having been placed in an auction setting into disease categories. Additionally, in lines 563-571, we discuss how our BRD diagnosis approach is one of several types of visual assessment schemes, and we may have misclassified cattle having subclinical disease prior to diagnosis.
  • To address the reviewer’s comments related to frequency of BRD in this population, we have included lines 522-530. While we agree that the disease rate in this population was high, especially given the relatively low-risk environment in which they were transported from, it is unknown at this time what specific features most contributed to this rate of disease. Very few published studies exist evaluating whole-life characteristics of cattle in respect to BRD morbidity and regarding cow-calf phase attributes that may correlate with BRD in later production cycles. Moreover, this study is part of an ongoing three-year trial to determine how preweaning vaccination and marketing decisions related to BRD morbidity and mortality throughout the course of life. Information gained from the subsequent two trials remaining for this overarching study, which are ongoing, may eventually help explain the rate of disease post-shipment in these specific cattle.

Reviewer 2 Report

The submission has as a goal "Our objective was to compare the influence of marketing on host transcriptomes measured on arrival at a backgrounding facility on the subsequent probability of being treated for BRD during a 45-day backgrounding phase".  Overall in many aspects the results sounds great. However, the cost of this procedure was not taken into acount. During background there is many aspects beyond the transcriptome that have an increase of respiratory disorders. However, it is very important to have some genetic informations..what it will be cost? I  my opinion, the economic can not be dissociated from the final price of the animal product. This must be addressed deeply in this study.

Author Response

  • Regarding the cost of our procedure and price of the animal product, we believe the reviewer is discussing the total cost of transcriptome evaluation and/or the phrase “marketing”. Our overarching study objective was to experimentally evaluate the whole blood transcriptomes of these cattle in an effort to determine differentially expressed genes which may be later used to predict BRD in similar populations and/or production settings. As such, we did not include the costs of the analysis as that was beyond the scope of this study. The term “marketing” is a colloquial term used in the cattle production industry to illustrate the program and, loosely, clinical history of newly-arrived cattle. While there are several purchasing programs used in North American beef production, such as off-farm selling, replacement herds, and sale barn purchasing, the two systems simulated by our study are very commonly employed.
  • Regarding the aspects beyond the transcriptome related to respiratory disorders, including that of genetic information, we believe the reviewer is referring to limitations of disease and experimental assessment with our study. We agree there are limitations to this study; one such limitation, addressed in lines 506-518, is the lack of pathogen, specifically viral, information that may further delineate disease outcomes and possible transmission patterns if evaluated in a time-course manner. Another is related to the timing and frequency of disease in this population, which is detailed on lines 522-530 and 555-558. While sampling over several time points may have uncovered transcriptomic information that could further elucidate genes and mechanisms for BRD during the background period, the central objective of this study was to evaluate at-arrival samples for features that would potentially predict marketing strategy and/or disease outcome. Additionally, we have included language to lines 512-514 to address the elements of genetic information, as the genomic mechanisms responsible for these gene expression patterns is not well understood with regard to cattle populations moving through these production systems.